# Investigation of an Explosion at a Styrene Plant with Alkylation Reactor Feed Furnace

**Yao-Chang Wu [1], Bin Laiwang [1] and Chi-Min Shu [1,2,*]**

[1]    Graduate School of Engineering Science and Technology, National Yunlin University of Science and Technology, Yunlin 64002, Taiwan; wuuyj5437@yahoo.com.tw (Y.-C.W.); anderson44005618@gmail.com (B.L.)

[2]    Center for Process Safety and Industrial Disaster Prevention, School of Engineering, National Yunlin University of Science and Technology, Yunlin 64002, Taiwan

*    Correspondence: shucm@yuntech.edu.tw; Tel.: +886-5-534-2601 (ext. 4416)

**Abstract:** To prevent and mitigate chemical risks in the petrochemical industry, such as fires and spillage, process safety management (PSM), is essential, especially where flammable, corrosive, explosive, toxic, or otherwise dangerous chemicals are used. We investigated process safety (PS) between man–machine (material equipment) and environmental interfaces by using process hazard analysis (PHA) and fault tree analysis (FTA). By analyzing the data obtained through machinery and mechanical integrity (MI), pre-startup safety review (PSSR), current operating modes, and areal locations of hazardous atmospheres (ALOHA) simulations of the disaster's aftermath, the cause of the styrene plant accident was found to be the fuel furnace (F101) switching process. Although the furnace had been extinguished, fuel continued to enter the furnace, and it was exposed to a high-temperature surface, resulting in the flashing ignition of the C4 fuel. The plan-do-check-act (PDCA) management model can be used to forestall the system from accident, and it is used to improve the proposal and develop countermeasures that would increase PSM performance and substantially lessen the impact of the thermal hazard. Disasters are often attributable to the unsafe state of machinery, equipment, or the environment, dangerous behaviors of the operator, and the lack of a thorough management system. It is anticipated that the investigation and analysis of the accident would not only find the real cause of the disaster but also lead to the establishment of better effective solutions for common safety problems.

**Keywords:** process safety management; pre-startup safety review; simulation; aftermath; effective solution

## 1. Introduction

Although the petrochemical industry has brought convenience to modern life, some accidents sporadically occur during the production process. In general, the process of raw materials poses enormous threats to human life. If improperly handled in preparation operation, management, transportation, or storage, these chemicals can cause fires, explosions, or leakages of toxic gas. In addition to causing loss of life and property damage, these kinds of industrial accidents could also trigger public protest over these large-scale plants. Therefore, after complex disasters happen, the social cost is often difficult to estimate. Nevertheless, the impact of a recent disaster was clear. On the morning of 6 March 2017, an explosion thundered through a styrene monomer (SM) company in Taiwan. In the aftermath of this incident, the public safety and environmental quality in the vicinity of industrial park were seriously affected. In light of the growing frequency of industrial accidents, this paper and its subject served as a relevant case study in the critical importance of process safety management (PSM). We hope to assist manufacturers in dealing with their accident investigation

reports and to help experts and scholars diagnose the causes of disasters to ensure that similar accidents will not occur in the future. In this way, the lives of employees and the property of manufacturers could be protected. Taiwan's SM plant alkylation reactor feed heating furnace explosion (equipment number: F101) accident on 6 March 2017 has been used as an example of PSM to provide analysis information to substantially lessen the risks as well as severities to the petrochemical industry [1,2].

In chemical manufacturing, the heating furnace is a process equipment that exchanges heat energy in furnace to heat the feed fluid to specific temperature range. The heating furnace is the main unit of the chemical processing system, fluid flows through the tubes into the furnace. Outside the tube is the heat generated by the combustion of fuel and air, so that the fluid is heated to a specific temperature. After the fuel is ignited by a burner, a flame is generated, and the combustion air flowing in the furnace is heated to exceptionally high temperature. Then, the combustion gas is transferred to heat the pipe by means of radiation and convection; the fluid is heated to the setpoint of the temperature for the process. Generally speaking, the gas in the heating furnace is not only extremely flammable, explosive, but also corrosive and toxic. Since the furnace is operated under high temperature conditions, the operating conditions may be poor; the gas generation period is short. If there is negligence or violation of the operating procedures, it may cause a fire or an explosion.

The heating of raw materials in chemical plants mainly depends on the heating furnace. The furnace temperature is sometimes as high as 700–800 °C (or even higher). Once the material of the furnace tube leaks, whether the temperature exceeds the self-ignition point or not, it is directly in contact with the open flame, so the hazard is extremely serious. In addition, improper ignition could also cause explosion. Once the furnace is damaged, it will directly lead to downtime and increase the loss. Therefore, it is necessary to take measures to prevent and mitigate fire and explosion accident.

Before commencing with the study and investigation, a brief literature [3] search was implemented to collect publicly available information about year's explosion accidents in furnaces. Tables 2 and 3 report this information. It was observed that even with detailed design, these accidents still occur; because they occurred at a high pressure within the system, they result in the loss of property and life. The accidents reported in Tables 2 and 3 indicate that most of them are caused by human error, improper maintenance, or faulty design. Accordingly, it is recognized that the accidents of heating furnaces occur sporadically. Common accidents mainly include furnace explosion, coking in the furnace, and explosion in the flue. The furnace tube is damaged, the material in the tube leaks into the furnace, fire occurs, and the fire or explosion caused by overheating. In addition, the burner of heating furnace is damaged, the refractory insulating material in the furnace is impaired. The environment is seriously polluted, and the heating furnace has exploded. In numerous literature descriptions, furnace explosion was the most common, destructive, and extremely serious accident. In the past, there have been few literature studies on accident cases. Therefore, we used the furnace explosion accident as an example to analyze and improve the understanding of the explosion accident of the heating furnace.

**Table 1.** Furnace explosion accident case summary in U.S.

| Date of Incident | Location | Specific Details | Explosion Details | | Injuries/ Fatalities |
|---|---|---|---|---|---|
| | | | **Cause** | **Consequence** | |
| 2 November 2002 | Vernon, CA | Used for smelting lead | Weakened furnace wall (holes) compromised the integrity of the furnace | Explosion spewing hot lead slag and dust | 1 employee received 2nd and 3rd degree burns |
| 16 March 2004 | Cucamonga, CA | Electric arc furnace | Attempted maintenance on a water leak from the furnace, loud popping noise resulted in explosion | Explosion resulting in hot steam and flying debris | 1 severe injury, 2 minor burns and cuts |
| 10 March 2006 | Midlothian, TX | Melt shop furnace (steel) | Furnace was tilted forward to begin tapping when the hydraulic hose cylinder failed | Molten steel leaked out of slag door, slag pot overfilled and an explosion occurred | 1 employee was killed |

**Table 2.** Furnace explosion accident case summary in U.S.

| Date of Incident | Location | Specific Details | Explosion Details | | Injuries/ Fatalities |
|---|---|---|---|---|---|
| | | | **Cause** | **Consequence** | |
| 27 May 2007 | Coatesville, PA | Electric arc furnace | Molten steel caused a water leak to become superheated high-pressure vapor | Explosion, molten steel | 1 employee killed, 2 seriously burned |
| 29 November 2007 | Manchester, GA | Used to melt aluminum | Aluminum car rims were placed into the furnace, moisture was still on the rims and a violent explosion occurred | Explosion, molten metal | 1 fatality, 6 serious injuries |
| 21 March 2011 | Louisville, KY | Large electric arc furnace | Water leaked into furnace which caused an overpressure event | Explosion from overpressure that sent furnace contents spewing into air | 2 workers killed, 2 seriously injured |
| 21 September 2014 | Fairfield, AL | N/A | Workers were opening/closing a furnace valve that contained oxygen and hydrated lime, while the furnace was in operation | Fiery explosion | 2 workers killed, 1 critically injured |

**Table 3.** Select cases on furnace explosion accident in Taiwan and Mainland China.

| Date of Incident | Location | Equipment | Explosion Details | Injuries/Fatalities |
|---|---|---|---|---|
| 7 November 2006 | Mainland China | Furnace | During fuel gas into the furnace, causing the gas pipe of heating furnace to burst and explosion. | Equipment damage |
| 3 May 2008 | Mainland China | Furnace | Heating furnace tube coking. | Equipment damage |
| 6 March 2017 | Taiwan SM plant | Furnace | In the diesel plant, during starting the furnace, the tube of furnace was cracked and led to explosion. | 4 seriously injured |
| 29 January 2018 | Taiwan Taoyuan Refinery | Furnace | In the diesel plant, during starting the furnace, the tube of furnace was cracked and lead to explosion. | No casualties Equipment damage |

We integrated the furnace explosion case and analyzed the relevant factors comprehensively that may cause fire and explosion during the operation of the furnace. The explosion factors of the heating furnace system were established by fault tree analysis method, and qualitative analysis was carried out to provide a basis for diagnosis and prevention of furnace explosion. The results of this study could provide the operators and managers with further understanding of the elements of furnace explosion and substantially diminish the chance of accidents in the future.

## 2. Accident Review

### 2.1. Accident Description

At 03:00 a.m. of 6 March 2017, one operator exchanged fuel gas for fuel oil in the alkylation reactor feed heating furnace (F101) and found poor combustion occurring at 04:28 a.m. The operator began to adjust accordingly, but at 04:31 a.m., a flash explosion occurred in the heating furnace, injuring four operators. It also damaged the surrounding property, such as the ethylbenzene process area (EB1) heating furnace (F101) being destroyed, both of which had to be shut down and repaired [4,5]. The accident scene conditions are shown in Figure 1; the location of the accident in Figure 2. The heating furnace (F101) accident's time sequence is shown in Figure 3 [6].

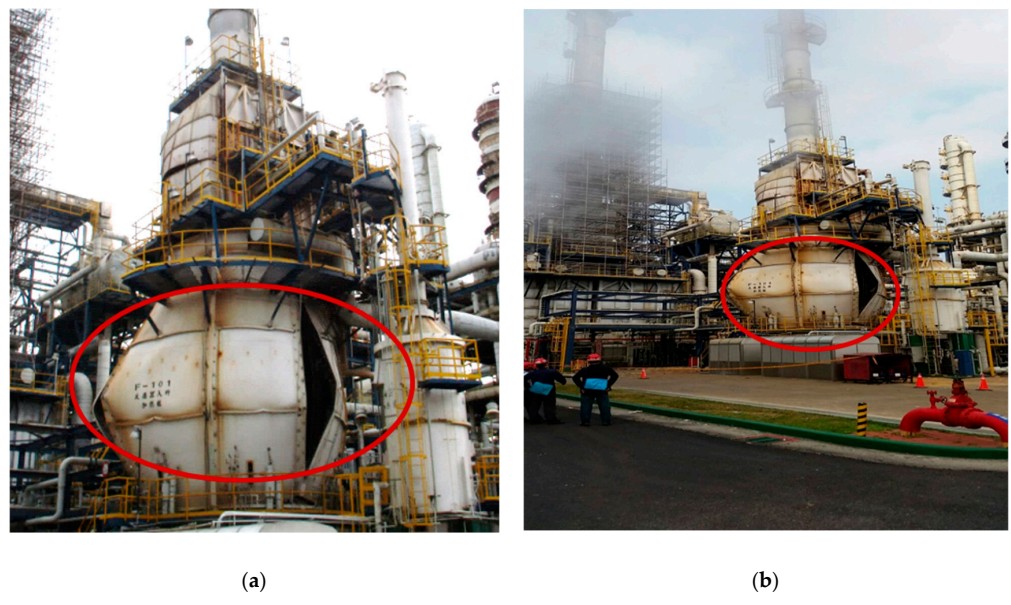

(**a**)                  (**b**)

**Figure 1.** Accident scene photos: (**a**) Heating furnace (F101) shell cracking situation-part 1 and (**b**) Heating furnace (F101) shell cracking situation-part 2.

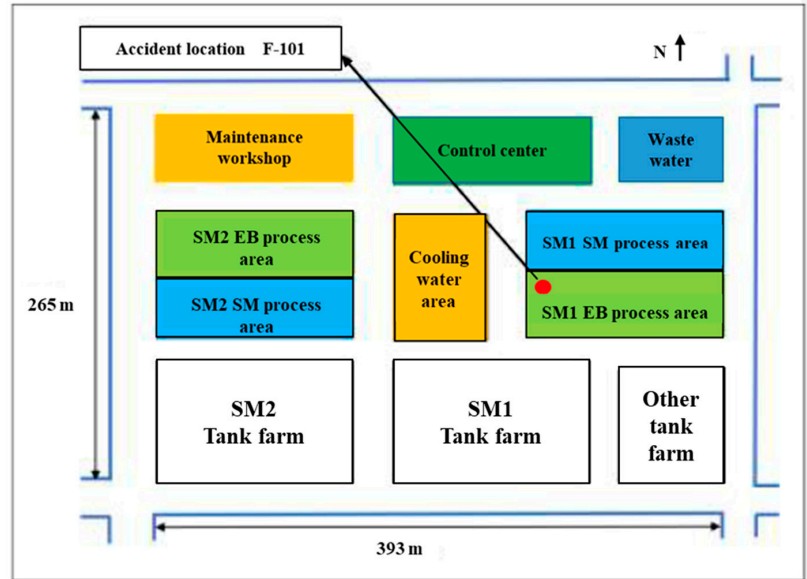

**Figure 2.** Location of the accident site.

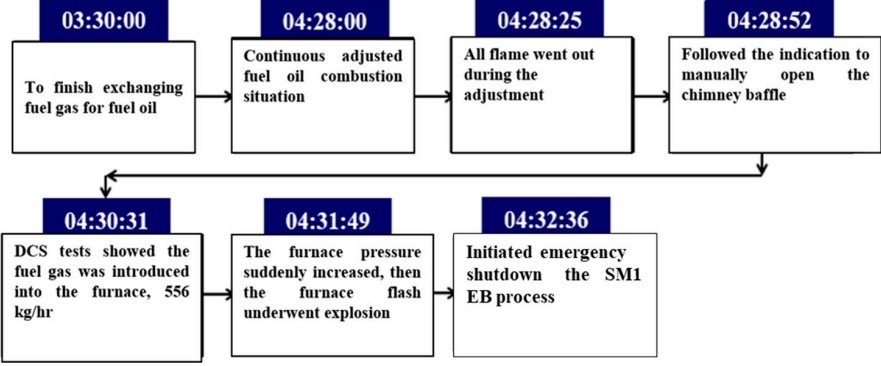

**Figure 3.** Time sequence for the heating furnace (F101) accident. DCS: distributed control system; EB: ethylbenzene; SM: styrene monomer.

## 2.2. Process Description

In general, the styrene monomer (SM) manufacturing method is based on the alkylation of benzene and ethylene. Ethylbenzene is formed and then dehydrogenated to produce a product–styrene, with byproducts: Toluene and hydrogen [7] (Figure 4). According to the process flow chart, before the raw materials enter the reactor, the materials (benzene) must be heated to reach the operating temperature by the heating furnace (F101). The inner structure of heating furnace (F101) is shown in Figure 5.

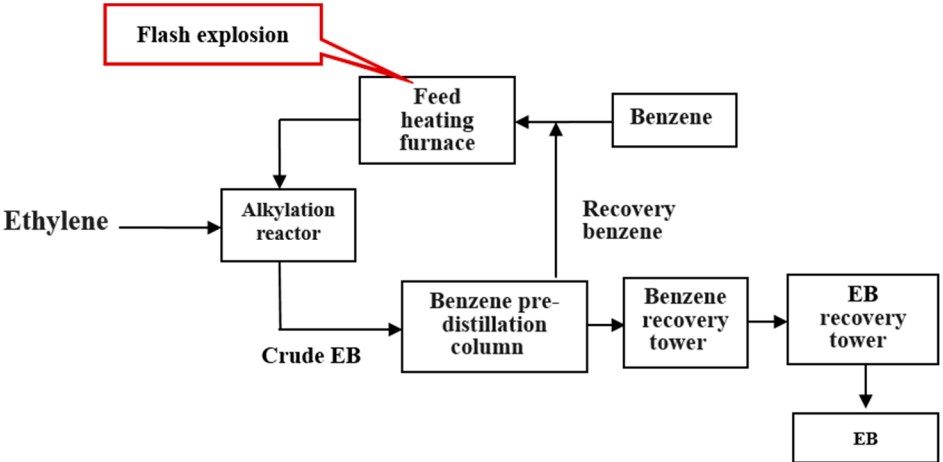

**Figure 4.** Process diagram and reaction equation for the alkylation reaction.

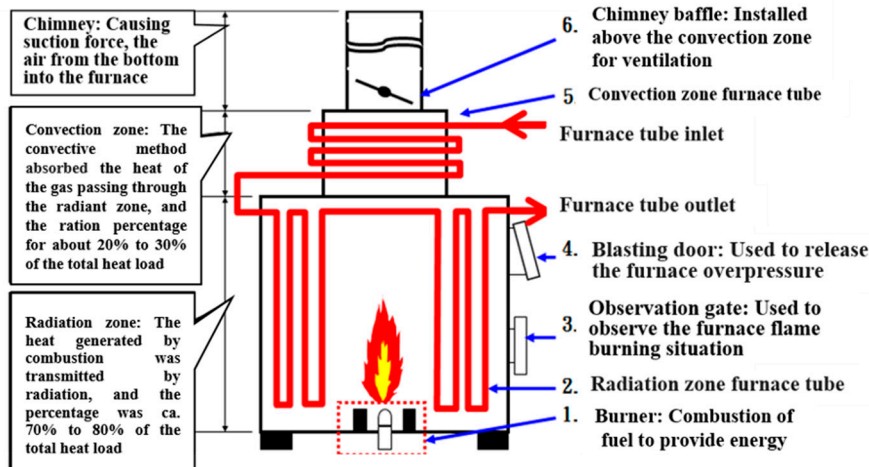

**Figure 5.** Schematic diagram of F101 furnace.

SM is an important intermediate material for the petrochemical industry that can be used to produce polystyrene (PS) plastic, acrylonitrile-butadiene-styrene (ABS) copolymers, styrene butadiene rubber (SBR), and unsaturated polyester (UPS). Its products are commonly used in electrical, mechanical, electronic, automotive, packaging, and other industries, all closely related to peoples' livelihoods [8].

The accident was a section of the EB process. The raw material benzene was heated by a heating furnace (F101), and then mixed to reaction with another raw material ethylene in the alkylation reactor. The design of the heating furnace (F101) fuel operating system included fuel oil and fuel gas functions. However, the furnace flash explosion occurred when fuel oil and fuel gas were switched [9].

## 3. Accident Investigation Analysis

### 3.1. Analysis Procedures

This section identifies and describes the accident which caused with current risk management methods and principles. According to the risk assessment process, including risk analysis, control, and four decision making steps [10], the implementation steps are as follows (Figure 6).

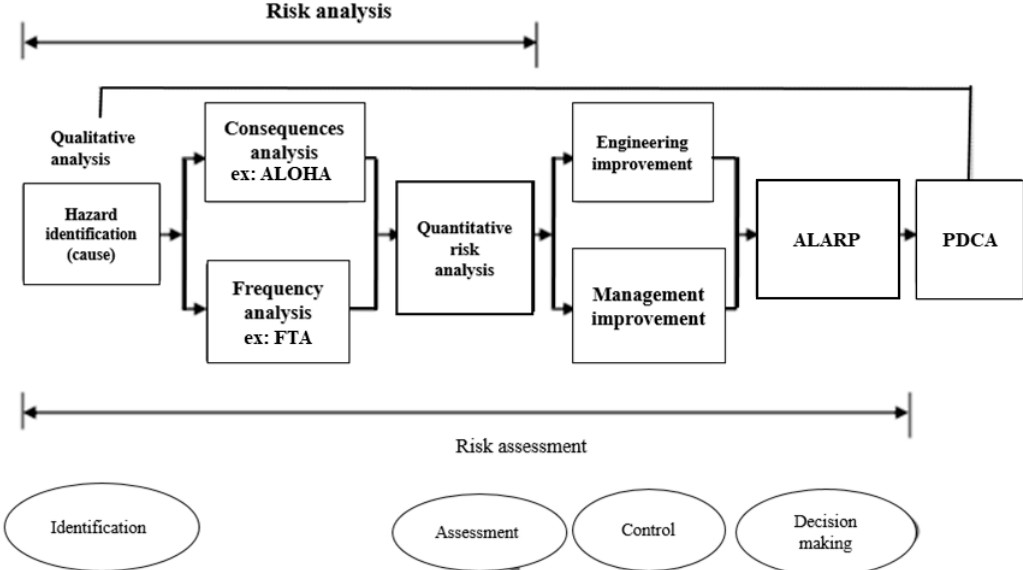

**Figure 6.** Accident case risk assessment process [10]. ALOHA: areal locations of hazardous atmospheres; FTA: fault tree analysis; PDCA: plan-do-check-act; ALARP: risks reduced to levels that are as low as reasonably practicable.

The accident analysis follows a four-step sequence:

- Hazard identification: First, identify all process activity and find the accident hazard causes and consequences, then confirm existing protective measures.
- Assessment: Use qualitative or quantitative analysis method to implement risk assessment.
- Risk control: According to the risk assessment results compare with the critical risk value, then take control measures for risk reduction.
- Decision making: According to the risk reduction control measures and principles, consider the cost-effectiveness to formulate improvement strategies, and regularly implement supervision and assessment to determine the risk reduction performance [11–14].

### 3.2. Fault Tree Analysis (FTA)

#### 3.2.1. What is FTA

FTA is one of the most widely used methods in system reliability, maintainability, and safety analysis. FTA was developed by Watson, at Bell Telephone Laboratories, in 1962 to apply the analytical method of studying the causes of undesired event failures. As a false logic diagram, the graphical structure of the model could be used as qualitative analysis and quantitative analysis [15].

The main purpose of the FTA is to identify potential causes of system failures before the failures actually occur. It also can be used to evaluate the probability of the top event using analytical or statistical methods. These calculations involve system quantitative reliability and maintainability information, such as failure probability, failure rate, and repair rate. After completing an FTA, the focus is on efforts for improving system safety and reliability. An FTA begins by defining the "undesirable"

top event, such as fire disaster, explosion, leakage, and out of control. Then, FTA can analyze the top event to basic events through intermediate events. Once the FTA has been constructed, it can be qualified and quantified [16,17].

### 3.2.2. FTA Effects

Effects of FTA are illustrated as follows:

- Applies deduction methods to figure out the possible cause of the system failure.
- Provides a clear graphical method, various easy ways to understand, and to count system failure.
- Points out the weaker links of the operating system.
- Renders system tools to evaluate system improvement strategies.

### 3.2.3. Fault Tree Establishment

In this study, the disaster investigation team summarized the accident investigation report for the company and analyzed the causes of the accident by FTA to find numerous points of the accident. The establishment of FTA is shown in Figure 7. The top event was a heating furnace (F101) explosion. The heating furnace (F101) exploded and the fuel continued to enter and accumulate in the furnace. Then, the fuel and air formed a mixed flammable substance (here, concentration was already within the boundaries of the explosion) that was exposed to the furnace surface caused by high temperature flashover (with enough energy) [18,19].

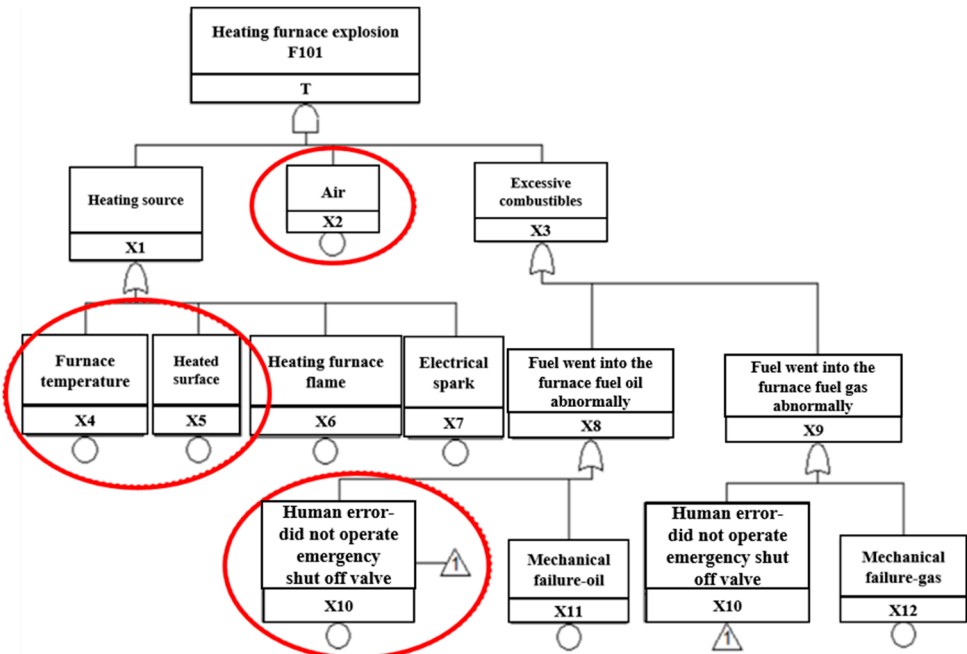

**Figure 7.** FTA diagram for this study.

Factors of the heating furnace explosion by FTA are as shown in Table 4. We discussed the causes of heating furnace explosion disasters from the principle of burning triangles, including three elements, fuel (flammable substances), oxygen (air), and heat (temperature). Then, we applied the FTA method to find the reasons for the explosion accident; the meanings of the symbols are as follows:

T Heating furnace explosion: The result of the furnace accident

X1    Heating source: One of the possible causes of the furnace accident
X2    Air: One of the possible causes of the furnace accident
X3    Excessive combustibles: One of the possible causes of the furnace accident

X4   Furnace temperature: One of the possible causes of heating source

X5   Heated surface: One of the possible causes of heating source

X6   Heating furnace flame: One of the possible causes of heating source

X7   Electrical spark: One of the possible causes of heating source

X8   Abnormal fuel flow into the furnace: Excessive combustibles (oil) in the furnace

X9   Abnormal gas flow into the furnace: Excessive combustibles (gas) in the furnace

X10  Human-error: Did not operate emergency shutoff valve.

Indirect possible reasons are as the following:

- After the heating furnace (F101) explosion, the operator did not shut down the F101 according to the standard operating procedure (SOP); then the atomization of fuel and gas continued into the furnace, resulting in flash over under the high temperature.
- In the fuel adjusting period, the furnace produced black smoke. At this point, on-site staff discovered the flame extinguished and performed the adjustment, then determined the fuel gasification and caused the ignition, resulting in other burners flame extinguishing instantly.
- The on-site staff response time during the abnormal situation was too short. The staff lacked awareness of the safety of the heating furnace operation and was unwilling to shut down the heating furnace (F101), nor did the staff perform the emergency stop procedure.
- Another reason might be that the staff was unfamiliar with the operating environment and equipment.

**Table 4.** Factors table of the heating furnace explosion (F101) fault tree.

| Factor | Meaning | Factor | Meaning |
|--------|---------|--------|---------|
| T | Heating furnace explosion | X1 | Heating source |
| X2 | Air | X3 | Excessive combustibles |
| X4 | Furnace temperature | X5 | Heated surface |
| X6 | Heating furnace flame | X7 | Electrical spark |
| X8 | Into the furnace fuel oil abnormally | X9 | Into the furnace fuel gas abnormally |
| X10 | Human errors—do not operate emergency shutoff valve | X11 | Mechanical failure—oil |
| X12 | Mechanical failure—gas | | |

*3.3. Qualitative Analysis*

The goal of fault tree analysis was to find the basic cause of the accident. That is, the basic events were found by Boolean algebra simplified calculation to obtain the minimal path sets, minimal cut sets (MCSs) of fault tree and the structure importance coefficient of each basic event. To reduce the probability of an accident, we could apply the consequence of analysis to take more preventive and control measures [20–23].

The fault tree totally included top event equations illustrated as below:

$$T = X1 \cdot X2 \cdot X3 \tag{1}$$

$$T = (X4 + X5 + X6 + X7) \cdot \cdot X2 \cdot (X8 + X9) \tag{2}$$

$$T = X2 \cdot (X4 + X5 + X6 + X7) \cdot \cdot (X10 + X11 + X10 + X12) \tag{3}$$

$$T = X2 \cdot (X4 + X5 + X6 + X7) \cdot \cdot (X10 + X11 + X12) \tag{4}$$

$$
\begin{aligned}
T = {} & X2 \cdot X4 \cdot X10 + X2 \cdot X4 \cdot X11 + X2 \cdot X4 \cdot X12 + X2 \cdot X5 \cdot X10 + X2 \cdot X5 \cdot X11 \\
& + X2 \cdot X5 \cdot X12 + X2 \cdot X6 \cdot X10 + X2 \cdot X6 \cdot X11 + X2 \cdot X6 \\
& X12 + X2 \cdot X7 \cdot X10 + X2 \cdot X7 \cdot X11 + X2 \cdot X7 \cdot X12
\end{aligned}
\tag{5}
$$

where T is the top event; the intermediate events are X1, X3, X8, and X9; basic events are X2, X4, X5, X6, X7, X10, X11, and X12.

MCS is a combination of basic events which would lead to the top events occurring. It cannot be simplified again and still ensure the occurrence of the top events. If this MCS is in the fail state, the entire system is out of order state.

$$K_1 = [X2\ X4\ X10],\ K_2 = [X2\ X4\ X11],\ K_3 = [X2\ X4\ X12] \tag{6}$$

$$K_4 = [X2\ X5\ X10],\ K_5 = [X2\ X5\ X11],\ K_6 = [X2\ X5\ X12] \tag{7}$$

$$K_7 = [X2\ X6\ X10],\ K_8 = [X2\ X6\ X11],\ K_9 = [X2\ X6\ X12] \tag{8}$$

$$K_{10} = [X2\ X7\ X10],\ K_{11} = [X2\ X7\ X11],\ K_{12} = [X2\ X7\ X12] \tag{9}$$

From the above result of analysis, we could find the key factors of top event are air, any heating source, and human error or mechanical failure.

### 3.3.1. Importance (I) of the Basic Events

We performed a relative importance sorting of the base events and the MCSs. The purpose was to find the basic events and MCS that contributed more to the top event. According to risk management, if the basic event and the MCS contributed much to the top event, we proposed to lessen the probability of the risk. The risk reduction would be more prominent. Based on the cost-effective consideration, we established the priority time ranking for completing the improvement suggestions.

In general, an MCS containing which has only one basic event was more likely to cause system failure than an MCS which contained two basic events. An MCS containing two basic events was easier to make a system malfunction than a minimum cut which contained three basic events. By analogy, the fewer basic events the MCS has, the more likely the system would fail. From the MCS, we can acquire the basic events importance sorting (Table 5). According to Table 5, the importance of the basic event is calculated and sorted as follows:

$$I(X2) > I(X10) = I(X11) = I(X12) > I(X4) = I(X5) = I(X6) = I(X7) \tag{10}$$

**Table 5.** Basic events importance sorting. MCS: minimal cut sets.

| Basic Event Code | The Times of Occurrences in the MCS | Importance Sorting |
|:---:|:---:|:---:|
| X2 | 12 | 1 |
| X4 | 3 | 3 |
| X5 | 3 | 3 |
| X6 | 3 | 3 |
| X7 | 3 | 3 |
| X10 | 4 | 2 |
| X11 | 4 | 2 |
| X12 | 4 | 2 |

Importance sorting of the basic event provides the theoretical basis for preventing and controlling measures of the fire and explosion accident in heating furnace. From a system security perspective, to prevent the top event and promote the reliability of equipment, we could use the fault tree to convert into the corresponding success tree. The success tree could be, in turn, simplified by using Boolean algebra and obtaining the minimal path sets (MPSs):

The success tree includes:

$$T' = (X1 \cdot X2 \cdot X3)' \tag{11}$$

$$T' = X1' + X2' + X3' \tag{12}$$

$$T' = (X4+X5+X6+X7)' + X2' + (X10+X11+X12)' \tag{13}$$

$$T' = X4' \cdot X5' \cdot X6' \cdot X7' + X2' + X10' \cdot X11' \cdot X12' \tag{14}$$

$$P1 = [X2] \tag{15}$$

$$P2 = [X4, X5, X6, X7] \tag{16}$$

$$P3 = [X10, X11, X12] \tag{17}$$

Hence, the MPSs represent the degree of system safety. According to the above simplification results, there are three groups of MPSs, which means that three possible ways could help prevent the fire and explosion accident in the heating furnace. From the above analysis, we could find the key factors of system security (the furnace would not explode) were neither air in the furnace, nor any heating source and human error or mechanical failure.

From FTA and the success tree analysis, to prevent the explosion of the heating furnace and improve the reliability of the equipment, the following aspects could be considered to develop improvement measures.

- When abnormal status happens, reduce the air concentration in heating furnace by inert gas. Then, the previous installed emergency shutdown valve would cut off fuel gas from entering the heating furnace.
- Strengthen the management of heat sources, such as hot work and static electricity.
- Enhance the improvement of personnel operation capability and equipment reliability.

### 3.3.2. ALOHA Simulation Analysis

The results of ALOHA simulation analysis were immediately completed and provided for reference by the business authority. According to the simulation results, there are injuries within about 56 m and fatal risks within 10 m, as illustrated in Figure 8.

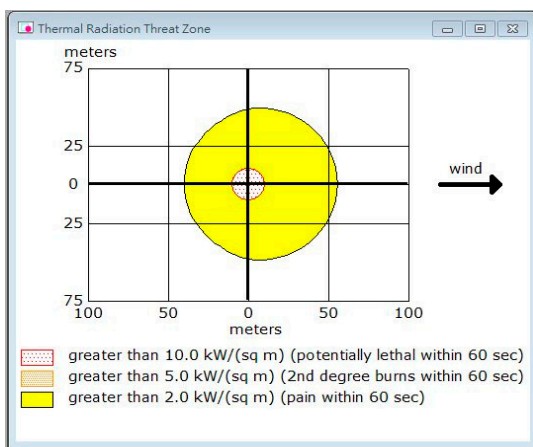 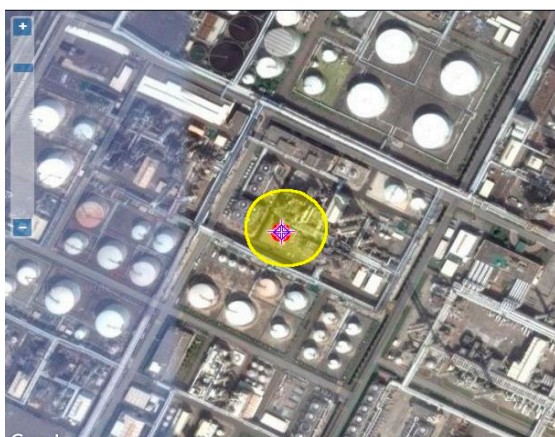

Influence region

(1)     Red: 10 m (10.0 kW/(m²) = potentially lethal within 60 s)

(2)     Orange: 10 m (5.0 kW/(m²) = 2nd degree burns within 60 sec)

(3)     Yellow: 56 m (2.0 kW/(m²) = pain within 60 sec)

**Figure 8.** ALOHA simulation results-C4 Gas for this study.

With reference to the simulation results, many people and equipment within the scope of influence could be considered to establish second management strategies and feasible emergency response mechanisms.

## 4. Results of Analysis and Discussion with the Method of Improvement

### 4.1. Results of Analysis

From the result of the qualitative analysis, this accident's causes could be figured out. Therefore, we could take measures to enhance the management, human behavior modes, operating procedures, equipment reliability, and other environmental conditions. The detailed methods of improvement are shown in Section 5.

It demonstrates a momentous point: When an abnormal operating situation occurred in the heating furnace (F101), after the F101 burner was extinguished, the fuel continued to enter the furnace and accumulated a high concentration of combustible particles. It was exposed to the high temperature surface of the furnace and caused flashing; the operator did not start the emergency interlock to close the fuel valve according to the SOP. Meanwhile, other conditions, such as air concentration and high temperature heat source, led to combustion and explosion conditions, so an accident occurred.

### 4.2. Improvement Strategy

Based on the accident investigation and qualitative analysis results, we recommended the following improvement (engineering or management) measures to prevent similar accidents.

#### 4.2.1. Improvement of Engineering Approach

A.    Mechanical and equipment integrity (MI)

- Remake the furnace body and check the status of the attached equipment as well as pipelines.
- Monitor equipment integrity: Record maintenance history and regularly replace equipment components.
- Add safety improvement advice, replace fuel gas cock valve with additional limited type switch, regulate all gas burners into the burner, as all could not be reset when the fuel valve was not fully closed.

B.    Inherent safety design (ISD)

To prevent furnace abnormal combustion (combustion gas short-circuit, incomplete combustion, and afterburning), we recommend installing a furnace combustion detection system and fuel gas interception interlock device.

According to PSM, 14 items were reviewed to make the ISD more secure. The device installation is shown in Figure 9:

(a)    Oxygen detector

- In the original design, there was only one oxygen detector. When the process was running, the offset anomaly could not be found immediately. One oxygen detector was added to improve the $O_2$ detection reliability of the equipment.
- Installed the other oxygen detector which was at the same height as the existing oxygen detector location; immediately notified the operator to maintain and calibrate it if the oxygen concentration difference was more than 1% original setting value.
- Checked the detector was vetted every month, which was using zero and full-scale calibration by standard gas.

(b)    Improvement of chimney baffle

- Made sure chimney baffle opening degree was controlled by distributed control system (DCS) in the original design. The signal was transmitted to the site controller to actuate the

chimney baffle. The DCS could not detect the actual opening degree value of the chimney baffle in the field.

- Installed a positioner in the chimney baffle and transfer the actual opening degree value of the chimney baffle on site to the DCS. Then, compared the value of DCS to master the opening degree with the actual value on site, whether DCS was consistent with the actual opening degree value of the site.
- Improved flame detector setting position.

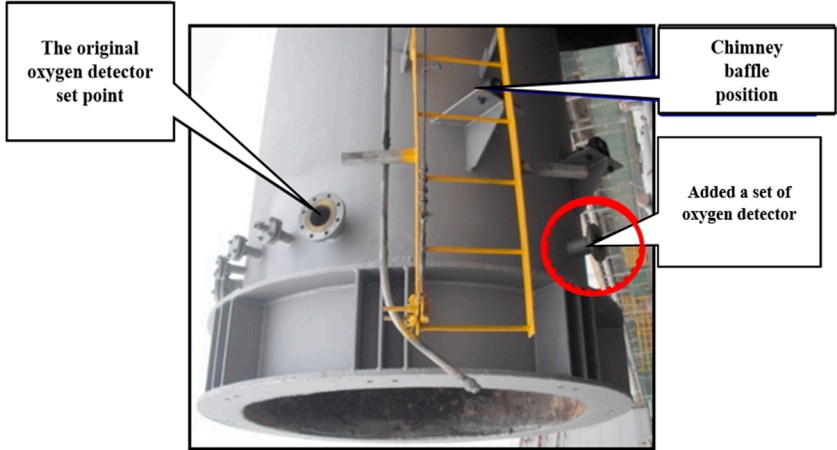

**Figure 9.** Strengthening inherent safety design (ISD) measures for the alkylation tower.

The flame detector scanned up from the bottom of the furnace in the original design. It was easy to accumulate ash in the detector lens surface and generate a false signal scanning. We recommended that the location of the flame detector could be changed to the side of the furnace wall; it scanned down to improve the case of false signals due to fouling. The flame detector setting position is shown in Figure 10.

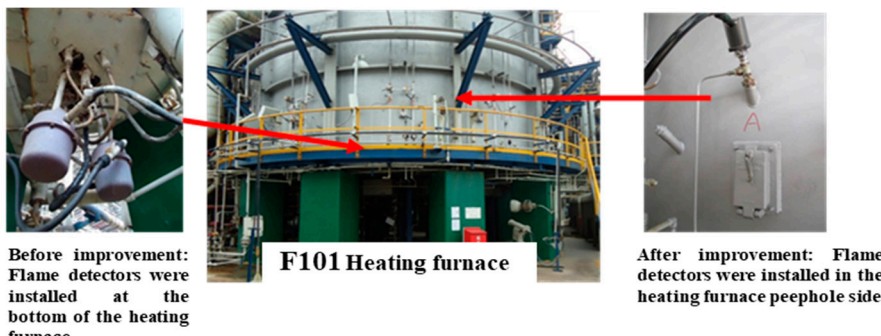

**Figure 10.** Design improvement of flame detector position.

4.2.2. Improvement of Management Approach

- Rechecked the entire plant furnaces SOP, strengthen operators' education and training.
- Established maintenance, check history, and check procedures.
- Pre-startup safety review (PSSR) for heating furnace (F101).

The PSSR is a systematic and thorough check to a process unit; the purpose of a PSSR is to help confirm that adequate safety measures are in place and are operational before a hazardous chemical is stored or into the process. Ensured that installations meet the original design criteria or operating procedure, then found any potential hazard and modify facilities to meet the management of change requirements. In other words, ensured the "Ready for Start-up" status of process facility/units.

The PSSR should confirm the following:

- Rechecked the PSSR of the heating furnace (F101). The PSSR covered four major aspects: Process, safety, and environmental protection (safety/fire protection/environmental protection), equipment (machinery) maintenance, and instrument equipment maintenance.
- Self-inspection by process and maintenance operators for the four major inspection items.
- A process hazard analysis for the heating furnace (F101). The recommendations should be implemented before startup, and modified facilities must meet the management of change requirements.
- Completed the training of each employee involved in operating a process.

4.2.3. Cost-effectiveness Analysis: As Low as Reasonably Practicable Sorting Analysis

If there is no industrial activity, it is completely free from risk. Many companies around the world ask that risks be reduced to levels that are as low as reasonably practicable, or "ALARP".

By definition, the ALARP principle is the minimum level of residual risk that needs to be reduced to reasonable limits. ALARP is used to reduce potentially harmful risks as much as possible. When deciding whether the ALARP point has been reached, the following factors should be seriously considered:

- Safety and health guide
- Codes of practice
- Industry practice
- Manufacturer specifications and government regulations
- Compliance with international standards and codes
- Comparison with dangerous events in similar industries
- Further reduction of the cost of risk in proportion to the benefits obtained.

A cost benefit analysis is used to determine if the risk has fallen to ALARP. This included weighing cost, time and technical sacrifices, and reducing risk levels. According to the above-mentioned investigation analysis and improvement measures (engineering method or management method), applying the cost effect analysis (ALARP principle) and sorting analysis (high risk and low cost) was proposed, and the priority was improved. The principles of ranking risk and cost are listed as Tables 6–9 [24,25]. The results of analysis are shown in Figure 11.

**Table 6.** Severity levels [26,27].

| Severity of Consequences | Description |
|---|---|
| 5 | *Safety*: One or more fatalities or permanent disabling injuries (PDIs) <br> *Environmental*: Major impact, making the national news <br> *Economic*: Losses greater than $10 million USD |
| 4 | *Safety*: PDIs, or serious injury to three or more people <br> *Environmental*: Continuous large impact, making the local news <br> *Economic*: Losses between $1 million and $10 million USD |
| 3 | *Safety*: Serious injury to one or two people, or minor injury to three or more <br> *Environmental*: Moderate impact, must be reported to environmental agency <br> *Economic*: Losses between $100,000 and $1 million USD |
| 2 | *Safety*: Minor injury to no more than two people. First aid. <br> *Environmental*: Minor impact <br> *Economic*: Losses less than $100,000 USD |
| 1 | *Safety*: No adverse health effects <br> *Environmental*: No detectable impact <br> *Economic*: Negligible economic impact |

**Table 7.** Likelihood of occurrence [26,27].

| Likelihood of Occurrence | Description |
|---|---|
| 5 | The event has happened several times at the plant. Likelihood is more than once in 1 year. |
| 4 | The event has occurred at the plant and frequently in industry. Likelihood is between once in 1 year and once in 10 years. |
| 3 | Incident has occurred at the plant, but is not common in industry. Likelihood is between once in 10 years and once in 100 years. |
| 2 | Incident has occurred in industry. Likelihood is less than once in 100 years |
| 1 | The event has a remote chance of happening and is unheard of in industry. |

**Table 8.** Classification of risk level [26,27].

| Frequency of Occurrence (Likelihood) | Consequences (Severity) | | | | |
|---|---|---|---|---|---|
| | Catastrophic (5) | Major (4) | Serious (3) | Minor (2) | Incidental (1) |
| Frequency (5) | 5 | 4 | 4 | 3 | 2 |
| Occasional (4) | 4 | 4 | 3 | 2 | 2 |
| Seldom (3) | 4 | 3 | 3 | 2 | 1 |
| Remote (2) | 3 | 3 | 2 | 1 | 1 |
| Unlikely (1) | 3 | 2 | 2 | r | 1 |

| Risk level | Risk Control Measures | Note |
|---|---|---|
| 5-extreme | Reduction risk measures need to be taken immediately, and tasks should not be started until the risk is reduced. | Unacceptable risk |
| 4-very high | Risk control measures must be taken within a certain period of time, and tasks cannot be started until the risk is reduced. | Unacceptable risk |
| 3-high | Based on cost or financial considerations, risk reduction measures should be taken gradually. | Unacceptable risk |
| 2-medium | There is no need to take risk reduction measures at the moment, but it is necessary to ensure the effectiveness of existing protection facilities. | Acceptable risk |
| 1-low | No risk reduction measures are required, but the effectiveness of existing safeguards must be ensured. | Acceptable risk |

**Table 9.** Classification of cost level [26,27].

| Cost Level | Description |
|---|---|
| E | Spending more than $10 million USD for risk reduction measures. |
| D | Spending range from $1 million to $10 million USD for risk reduction measures. |
| C | Spending range from $100,000 to $1 million USD for risk reduction measures. |
| B | Spending range from $10,000 to $100,000 USD for risk reduction measures. |
| A | Spending less than $10,000 USD for risk reduction measures. |

Where risk level is from 1 to 5 in the vertical axis. Numbers 1 to 2 are acceptable risk. Numbers 3 to 5 are unacceptable risk. Cost level is from A to E in the horizontal axis. The cost is gradually increasing from A to E. Numbers 1 to 7 are the order of reduction risk measures in the matrix. This is a risk decision matrix, representing the relationship between cost and risk, numbers 6 to 7, We can consider negligible or defer implementation. However, numbers 1 to 3 are listed as priorities for improvement and numbers 4 to 5 as an ALARP.

| Cost level | | | | | | | |
|---|---|---|---|---|---|---|---|
| | | **High** | | **Low** | | | |
| | | | E | D | C | B | A |
| **Risk** | **High** | 5 | 5 | 4 | 3 | 2 | 1 |
| | | 4 | 6 | 5 | 4 | 3 | 2 |
| **level** | | 3 | 7 | 6 | 5 | 4 | 3 |
| | **Low** | 2 | 7 | 7 | 6 | 5 | 4 |
| | | 1 | 7 | 7 | 7 | 6 | 5 |

**Note: 1.** Priority level ≤ 4 listed as a major security aspect.

**Figure 11.** Results of the cost-effective ALARP sorting analysis [26,27].

### 4.2.4. ALARP Sorting

The improvement measures were ranked according to the principle of ALARP (high risk and low cost), the results were as follows:

- Rebuilt the heating furnace (F101) body and evaluated the damage status of the surrounding ancillary equipment and pipelines.
- Rechecked the furnace SOP.
- Improved worker education and training.
- Monitored equipment integrity: Recorded maintenance history and regularly replaced equipment components.
- Established maintenance, checked history, and followed procedures.
- Erected a gas feed interruption device.
- Installed a furnace combustion condition detection device.

### 5. Conclusions and Recommendations

We learned from the fault tree qualitative analysis and field investigation results that the accident was caused by high temperature sources, mixed flammable gas, and human factors. Among them, the human factors were the most important reason, such as the classification of human factors (i.e., misoperation, misjudgment, unmoved action, lack of training, or lack of skill). The key points of human factors were on how people interact with tasks, with machine/equipment/technologies, and with the environment, in order to comprehend and assess these interactions. The goals of human factors were to optimize human and system efficiency and effectiveness, operation safety, health, comfort, and quality of living. If the interface among human body, machine and environment were not properly deployed, accordingly, the problem of human error should be solved from the heating furnace design.

On the other hand, when human factors are not avoided, we should improve from existing management systems and equipment.

Through FTA, cost-effectiveness analysis, and on-site investigation of the accident, the following recommendations were drawn:

- Units with different specializations must cooperate with each other to investigate accidents: Previous accident investigation reports biased in fire expertise should include chemical process and other background experts to investigate. To find the real root cause of an accident, we should establish an investigation team by accident procedure, and then take control measures to avoid the accident from happening again.
- To improve the operation safety of the equipment, including ISD, mechanical and MI.
- To develop procedures of MOC (management of change), when the heating furnace (F101) uses different fuels.
- To avoid increasing the risk of exchanging fuels (fuel oil and fuel gas), we suggest using a single fuel.
- To enhance the operators' ability of response when the heating furnace (F101) abnormal situation occurs.
- To improve operator training and equipment regular maintenance.
- Before the heating furnace was opened, PSSR for F101 was divided into four major aspects: Process, (industrial safety/firefighting/environmental protection), equipment (mechanical) maintenance, and instrument as well as electrical section (safety) maintenance.
- Case exchanging experience should be promoted to the grassroots.
- Installing a system for nitrogen into the furnace to reduce the concentration of combustible gases.

**Author Contributions:** Y.-C.W. wrote this manuscript; B.L. provided idea; C.-M.S. rendered suggestions for the manuscripts. All authors supplied comments on this theme.

**Funding:** This research received no external funding.

**Acknowledgments:** Thanks to Wei-Cheng Lin for his valuable suggestions and help.

**Conflicts of Interest:** The authors declare no competing interests.

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
