# Peer review of "Investigation of an Explosion at a Styrene Plant with Alkylation Reactor Feed Furnace"

_applsci, doi:10.3390/app9030503_

Round 1

Reviewer 1 Report

I read this paper with interest. It has merit but I have found it quite unbalanced. To make it clearer, the accident, the approach and the results are presented in detail. However, the first part of the paper is very poor, especially the introduction, which looks like more an introduction to a technical manual rather than a research paper. To help reviewers improve their paper I would require that they address the following: - (major) The introduction is too short and not informative. What is the contribution of the paper and what is the new that brings to the literature is not discussed. Furthermore, the paper is not placed well in the literature and what is the gap that comes to fill in. - (major) There is no literature review at all. The authors do not provide any credit to previous works. - (major) The main conclusions are not given. In the last section, authors should give in bullets what are the main conclusions of this work. - (minor) There are a few typos in the text.

Author Response

1. It is described from page 3 and line 21 to page 4 and line 14 of the full revised article.

2. It is described on page 4 and line 15 to page 5 and line 5 for the literature on the previous work.

Second, Tables 3 and 4 are also provided as the previous work in this area.

3. It is described on page 25 and lines 2–13 of the main Conclusions and Recommendations.

4. We have reviewed all text to revise the typos, and the manuscript has polished by Dr. Frank Oreovicz (retired) from Purdue University.

Reviewer 2 Report

The study to analysis actual accident at a styrene plant and indicate safety measures based on analysis is interesting.  My comments are as follows,

Please add the description of Table 1. Current manuscript does not have the description of Table 1 at all.

Please describe the concrete purpose of the present study.

In line 197 of page 8, heat sources should be ignition source (?).

At least, reviewer does not understand the Fig. 11(colour and number) since the description of Fig.11 does not have much.  Please add the explanation about Fig. 11 in detail.  In addition, please describe the ranking rule of frequency and severity of the scenarios.   

I suppose “C” appears twice in the horizontal axis.  The first one should be “E”.(?)

Author Response

1.  We have discussed the causes of heating furnace explosion disasters from the principle of burning triangles, including three elements, fuel (flammable substances), oxygen (air), heat (temperature). Then, to apply FTA method, and to find the reasons of explosion accident, the meaning of the symbols are as follows:

T   Heating furnace explosion: The result of the furnace accident

X1    Heating source: One of the possible causes of the furnace accident

X2    Air: One of the possible causes of the furnace accident

X3    Excessive combustibles: One of the possible causes of the furnace accident

X4    Furnace temperature: One of the possible causes of heating source

X5    Heated surface: One of the possible causes of heating source

X6    Heating furnace flame: One of the possible causes of heating source

X7    Electrical spark: One of the possible causes of heating source

X8 Abnormal fuel flow into the furnace: Excessive combustibles (only for oil) in the furnace

X9 Abnormal gas flow into the furnace: Excessive combustibles (only for gas) in the furnace

X10  Human error: Did not operate emergency shutoff valve

X11  Mechanical Failure-oil: Shut off valve failure cannot be turned on

12     Mechanical Failure-gas: Shutoff valve failure cannot be turned on.

It is described on page 13 and lines 6–21.

2.  This paper has integrated the furnace explosion case and analyzed the relevant factors comprehensively that may cause fire and explosion during the operation of the furnace. The explosion factors of the heating furnace system were established by FTA method, and qualitative analysis was carried out to provide basis for diagnosis and prevention of furnace explosion. The results of this study could provide the operators, and managers with further understanding of the elements of furnace explosion and to substantially diminish the chance of accidents in the future. It is described on page 5 and lines 6–12.

3.  The heat source meaning is ignition sources, including heated surface, furnace (hearth) temperature, electrical spark, and heating furnace flame. (page 13 and lines 6–21)

4.  The principle of ranking risk and cost is shown in Tables 5 to 8.

The results of analysis are shown in Fig. 11. Risk level is from 1 to 5 in the vertical axis. Numbers 1 to 2 are acceptable risk. Numbers 3 to 5 are unacceptable risk. Cost level is from A to E in the horizontal axis. The cost is gradually increasing from A to E. Numbers 1 to 7 are the order of reduction risk measures in the matrix. It is described on page 23 and line 11 to page 24 and line 3.

5.  The first one is E in the horizontal axis. It is modified in Fig. 11.

Round 2

Reviewer 1 Report

Thanks for addressing my comments.